# Building Korean DMZ Metaverse Using a Web-Based Metaverse Platform

Sangsu Choi, Kajoong Yoon, Miae Kim, Jintak Yoo, Bonghyeon Lee, Inho Song and Jungyub Woo *

R&D Research Center, IGI Korea, Seoul 08376, Korea
* Correspondence: jungyub.woo@ontwins.com

**Abstract:** Metaverse is a compound word of "Meta" and "Universe", meaning a world that transcends reality, a new virtual world. Due to the COVID-19 pandemic, non-face-to-face further accelerated the activation of the Metaverse. The Metaverse has the attractiveness of arousing user interest, high service scalability, and the potential to combine multiple revenue models. Therefore, many industries are adopting the Metaverse. The Republic of Korea has been on a ceasefire, since 1953, after the Korean War. After the war, the DMZ was decided in an area of 2 km from north to south (total of 4 km), centered on the Military Demarcation Line (MDL). The DMZ area is the largest nature conservation area in Asia and a military area where public access is strictly restricted. The Ministry of Unification is trying to reduce tensions and establish peace in the DMZ area by raising the interest of Korean people, as well as people around the world, through the DMZ Metaverse application. The Korean government has its own e-Government framework and cloud operating environments. In order to adopt the Metaverse to Korean government agencies and operate them sustainably, it is essential to support the e-Government framework and the cloud environments. In this paper, the design features, architecture, models, and functions of OnTwins, a Metaverse platform serviceable in the Korean government IT operating environment, are elaborated in detail. The DMZ Metaverse public service built on the platform will also be introduced. Finally, after comparing it with other Metaverse platforms, future research directions are discussed.

**Keywords:** Metaverse; Web3D; digital twin; virtual world; VR/AR





## 1. Introduction

Metaverse is a compound word of "Meta", meaning virtual and transcendence, and "Universe", meaning the world. In 2007, the Acceleration Studies Foundation (ASF) classified the Metaverse into four types: Lifelogging, Augmented Reality, Virtual World, and Mirror World [1]. Lifelogging is a technology that captures, stores, and depicts everyday experiences and information about objects and people. Augmented Reality refers to an environment that interacts through virtual, overlapping objects expressed in 2D or 3D in real space. Virtual World is an alternative world built from digital data, which may be similar to reality or completely different from reality. Games are a representative virtual world. Mirror World refers to an informationally expanded virtual world that reflects the real world as realistically as possible. Google Earth and digital twins are representative examples. The Metaverse is rapidly developing based on the combination of these four types [2]. Big tech companies, such as Meta (formerly known as Facebook), Microsoft, Apple, Google, and NVIDIA, are leading the Metaverse base technology market, and they are conducting R&D based on astronomical investment to gain technological advantage. Big tech companies are focusing on securing original technologies such as VR/AR devices, 3D GPUs, rendering, and Metaverse platforms [3].

The advantages of the Metaverse are as follows [4]. First, it has an attraction that can arouse customer interest. Through real-time interaction between avatars, users can feel a sense of presence and immersion in a space similar to reality, and users can feel vicarious

pleasure through experiential activities in places that are difficult to visit. Second, the level of service scalability is high. The Metaverse has a strong community element in which mutual relationships between people are formed. As the number of loyal users increases, the number of visits and use times increase, various services can be additionally combined. Third, multiple revenue models can be combined. Advertisements and item sales can be applied. Recently, Non-Fungible Token (NFT) technology has also been applied [5].

Due to these advantages, the Metaverse is being actively applied in various fields, and Korean government agencies are also actively reviewing the introduction of the Metaverse to increase public interest and participation. The Korean government has established its own e-government framework and operating environment. Therefore, in order to apply the Metaverse to Korean government agencies, there is an essential constraint that it must be installed and serviced, in a dedicated cloud environment, based on the e-government framework.

In this study, a Metaverse platform developed based on the Korean e-government framework and WebGL that can support the Korean government cloud operating environment is elucidated, and the Korean DMZ Metaverse building story is introduced. The rest of the paper is organized as follows. Section 2 examines technology trends and application case studies related to the Metaverse platform. Section 3 describes the futures, architecture, and functions of the Metaverse platform in detail. Section 4 introduces and discusses cases applied to the DMZ *Goseong* area and compares the developed platform with other platforms. Section 5 describes the conclusion and future research.

## 2. Related Technologies and Research

Roblox, Zepeto, Gather, Minecraft, Animal Crossing, Horizon Workrooms, and ifland are well-known major Metaverse platforms that have been commercialized [6]. Looking at the major platforms released after 2000, the Sims is a life simulation game played in a virtual 3D space. The 3D avatar and the ability to customize the interior of the house have made it very popular. In 2001, the Nintendo game Animal Crossing was released. In 2020, U.S. President Joe Biden conducted an election campaign through this platform. Roblox entered beta in 2004 under the name DynaBlocks Beta and was then officially released in 2006. In a way that users create their own content, in addition to role-playing, users can play many games created by various users, such as FPS (First-Person Shooter), escaping, and racing. Roblox is considered as one of the most successful Metaverse platforms so far. More than 150 million monthly users and over 40 million daily users are using Roblox. In the U.S., 55% of Gen Zers have signed up for Roblox, spending an average of 2.6 h per day. This spending time is three times as long as YouTube and seven times as long as Facebook [7].

Minecraft, released in 2011, supports activities such as building, farming, hunting, and more in a 3D world where everything is square. On 5 May 2020, Children's Day in Korea, the Korean government built the Blue House into a Metaverse using Minecraft, providing users with a service to tour the Blue House and meet the President. In 2018, Naver, a Korean internet conglomerate, launched Zepeto, an AR avatar based Metaverse platform. After automatically creating a virtual character from a photo, users can freely customize the appearance. Zepeto is actively collaborating with luxury fashion brands and K-pop stars. It has grown in popularity with the addition of the functions to allow users to update characters and content. It currently has 300 million users worldwide [7]. In 2020, Facebook Horizon was released [8]. It is a VR social media that allows users to communicate with other users around the world by participating in the virtual world, Horizon, with their own avatars. In 2021, Horizon Workrooms for meetings was launched. In 2020, Gather, a video conferencing platform that supports Super Mario-like 8-bit pixel avatars, was launched. In addition to meetings, it supports various social interaction activities such as virtual offices, university campuses, weddings, and birthday parties. In 2021, Korean conglomerate SK Telecom launched ifland with functions similar to the existing Metaverse platforms.

Duan et al. [9] pointed out that, although companies are already preparing for the infinite potential of the Metaverse with bold investments, there is not enough discussion in

academia that will scientifically lead the Metaverse improvement. In academia, there are many cases of using commercialized Metaverse platforms for education and analyzing their effects. Meier et al. [10] developed a simulation of the urban environment using Roblox. As a result of an experiment on the sculptural heritage of the city of Santa Cruz de Tenerife, it was emphasized that the Metaverse increases students' understanding. Lee [7] explained that the transition to the Metaverse era, a new virtual convergence platform, is accelerating as the importance of online virtual space grows due to the COVID-19 pandemic. He investigated the Metaverse construction principle and detailed functions using Zepeto. He argued that the Metaverse would be actively applied in games, exhibition halls, theme parks, marketing, education, medicine, and media. McClure and Williams [11] explained that the COVID-19 pandemic made a distance learning approach essential. They had a class with 38 students using the Gather platform. The results showed that the students preferred the Metaverse platform to online systems such as MS Teams. Cipollone, Schifter, and Moffat [12] used Minecraft for their high school literature class. They explained that the Metaverse has the advantage of better understanding the concept and demonstrating the creativity of students. Fisher et al. [13] allowed users to naturally experience environmental protection activities and environmental conservation education using Animal Crossing. They argued that these experiences have the potential to be translated into actual environmental protection activities. They also argued that the experience on the Metaverse could provide a tremendous opportunity to deliver an environmental message to billions of people in the real world. Kim [14] tested the possibilities and limitations of art education using the ifland platform. Experimental participants showed highly pessimistic feedback in the first experiment because they had little experience of using the Metaverse. However, as the experiment was repeated, interest increased, and most of the participants responded positively to the playful approach methodology. Kim mentioned that the more Metaverse technology develops, the more effective the application of art education would be.

Instead of using the commercialized Metaverse platforms, studies were also conducted to build and use their own Metaverse for education. Games and the Metaverse that form social relationships and conduct economic activities in a virtual space have many properties in common. Therefore, game engines, such as Unity and Unreal engines, are the most used to build the Metaverse. Unity engine can be applied to various fields, such as animation, construction, and manufacturing, and it has strength in building a mobile virtual world. It is more suitable for game development with casual or cute elements. Roblox and ifland are representative Metaverse platforms developed with Unity. The Unreal engine uses high-quality graphics and time-rendering technology, making it more suitable for creating high-quality games such as mechanics, simulators, racing, and FPS.

Duan et al. [9] introduces a Metaverse prototype of a blockchain-based university campus, developed with Unity, and discussed its design and insights. Kim, Lee, and Choi [15] developed a Metaverse system running in the HMD environment using Unity for virtual learning. They have used this platform for university and graduate classes such as drone piloting, the Capstone project, and YouTube creator courses. Positive results were witnessed in terms of students' interest and understanding. Lee, Woo, and Yu [16] pointed out that video classes have limitations in replacing face-to-face classes. After developing a Unity-based aircraft maintenance simulation solution that supports the HMD environment, a study comparing the Metaverse environment and video classes was conducted. A survey was conducted to measure the training method, educational effectiveness, and knowledge acquisition and maintenance test, and it was shown that education in the Metaverse was effective. Zhang et al. [17] developed a 3D virtual Weft-knitting Engineering learning system using Unreal Engine 4. They argued that the proposed system improved the teaching effects, and learners were more satisfied with the class using realistic visualization.

Studies on the building of the immersive 3D-based digital twins and the Metaverse using Web Graphics Library (WebGL) have also been published [18–20]. WebGL is a low-level JavaScript API for rendering 2D and 3D graphics on the web. Based on OpenGL ES 2.0, it is drawn on top of the HTML5 Canvas element built into the browser engine. WebGL has

the following advantages: anyone can use it without royalties, it uses graphics hardware that supports rendering acceleration, it runs embedded in the web browser without a separate plug-in, and furthermore, it is easy to handle if you have OpenGL experience. JavaScript programming is possible. It is also available in mobile browsers. Unity also allows developers to publish content to JavaScript using the WebGL build option. The content is executed in a web browser using HTML5 and WebGL rendering APIs. However, it is difficult to say that it fully supports the Web. Three.js and Babylon.js, which are JavaScript 3D libraries that can easily use 3D programming technologies such as rendering, camera, and lighting based on WebGL technology, are also being used often. Most of the studies related to the Metaverse are studies that analyze the effect of application in a specific field, such as education, using a commercial Metaverse platform. AThere are also many studies on detailed techniques and algorithms related to the Metaverse and games. However, there is very limited research on Metaverse platform development itself. The next section describes, in detail, the technologies and methods related to the development of a Metaverse platform.

## 3. OnTwins: Web-Based Metaverse Platform

As mentioned earlier, the Metaverse platform, applicable to the Ministry of Unification, must be installed in the cloud operation center in Gwangju to enable public services and must be developed based on the e-Government standard framework. Therefore, web standards must be complied with, and 3D visualization must also be developed based on WebGL. This section describes the Metaverse platform, design characteristics, architecture, model, and functions of OnTwins.

### 3.1. Design Features

OnTwins has design features such as Web Standards, Microservices, Kubernetes, Data Intensive Design, Media franchise, Single Page Application (SPA), Server-Side Rendering (SSR), and Client-Side Rendering (CSR).

3.1.1. Web Standards, Cross Browsing, Web Accessibility

- OnTwins has been developed considering Web Standards, Cross Browsing, and Web Accessibility.
  - Web standards are open Internet standards approved by standardization organizations, such as World Web Consortium (W3C), European Computer Manufacturers Association (ECMA) International, the Internet Engineering Task Force (IETF), and Organization for the Advancement of Structured information Standards (OASIS). W3C, a representative web standardization organization, conducts web standardization activities centered on six domains: Architecture, Interaction, Technology and Society, Ubiquitous Web, Web Accessibility Initiative, and Quality Assurance.
  - Cross Browsing is a technique that uses standard web technology to create web pages using common elements so that they are not optimized or dependent on either one of the operating systems and browsers. Compliance with web standards is included in the concept of web interoperability, which means that the same result is obtained between operating systems and browsers when using a website.
  - Web accessibility refers to the standard for accessing the web so that all people, including the handicapped, can use it equally when browsing information using a PC or other devices. Although web accessibility is highly emphasized in the basic philosophy of the web, numerous websites are being created with accessibility ignored. The web must stand on the principle of reciprocity and equality where everyone shares information evenly from the beginning, and OnTwins abides by it.

### 3.1.2. Microservices Architecture

Microservices architecture is an architecture-based approach to building application. The way applications are segmented into core functions is what differentiates microservices from traditional monolithic approaches. Each function is called a service and can be built and deployed independently. This means that individual services can operate (or fail) without negatively impacting other services. Microservices architecture has advantages, such as shortened time to market, easy deployment, high scalability, convenient access, excellent recovery ability, and improved openness.

### 3.1.3. Kubernetes

Kubernetes is a portable, extensible, open-source platform for managing containerized workloads and services that facilitates both declarative configuration and automation. Kubernetes provides a framework for elastically running distributed systems, handles application scaling and failover, and provides deployment patterns.

### 3.1.4. Data Intensive Design

Data Intensive Design is a data-centric design method used to cope with the amount of data, the complexity of data, and the speed at which data changes. It uses a data type, DB, similar to NoSQL and has a design philosophy focusing on how to store and process data by utilizing technologies such as message queues, caches, search indexes, batch processing frameworks, and stream processing frameworks.

### 3.1.5. Media Franchise

Media franchise is a collection of related media that several derivative works are produced from an original creative work of fiction. It is important to normalize and standardize media content so that it can be used on various platforms, devices, and media. This strategy, often called One Source Multi-Use (OSMU), increases the added value of content, and it increases the value of other content or application at the same time. It is an especially major method for continuously attracting the attention of consumers who flow through various channels.

### 3.1.6. SPA (Single Page Application)

SPA refers to a web application or website that interacts with the user by dynamically rewriting the current page without loading a new page from the server. Unlike the traditional method, the SPA method calls all necessary codes, such as HTML, JavaScript, and CSS, into a single page and receives only the necessary data, according to the user's response, and reorganizes the page. In general, since only the parts that need to be changed according to the user's response are reconfigured, it has advantages of being efficient, simple distribution, and providing a unified user experience.

### 3.1.7. SSR and CSR

- SSR means rendering work on the server. In the existing method, when a user accesses a web page, the server makes a request for the page, and the server interprets how resources, such as HTML and view, would look, renders it, and returns it to the user.
- CSR is a method in which the server first loads and displays the entire page, and then, whenever a user's request comes, the server provides the resource. Then, the client interprets and renders it. OnTwins supports both full SSR and full CSR, depending on the site and application characteristics. In particular, it supports dehydration-type SSR and pre-rendering-type CRS, which are more flexible methods.

### 3.2. Platform Development

#### 3.2.1. Architecture

Figure 1 shows the platform architecture. This platform consists of Frontend and Backend connected to DB/File Sever. DB/File Server manages data collected through

public data portals and self-produced data. Frontend is divided into an administrator mode that uses a builder to arrange and modify objects, as well as manage users and contents, and a user mode that enjoys the Metaverse space. The platform is developed based on the MVC pattern, and related standard technologies used are summarized and shown in Table 1.

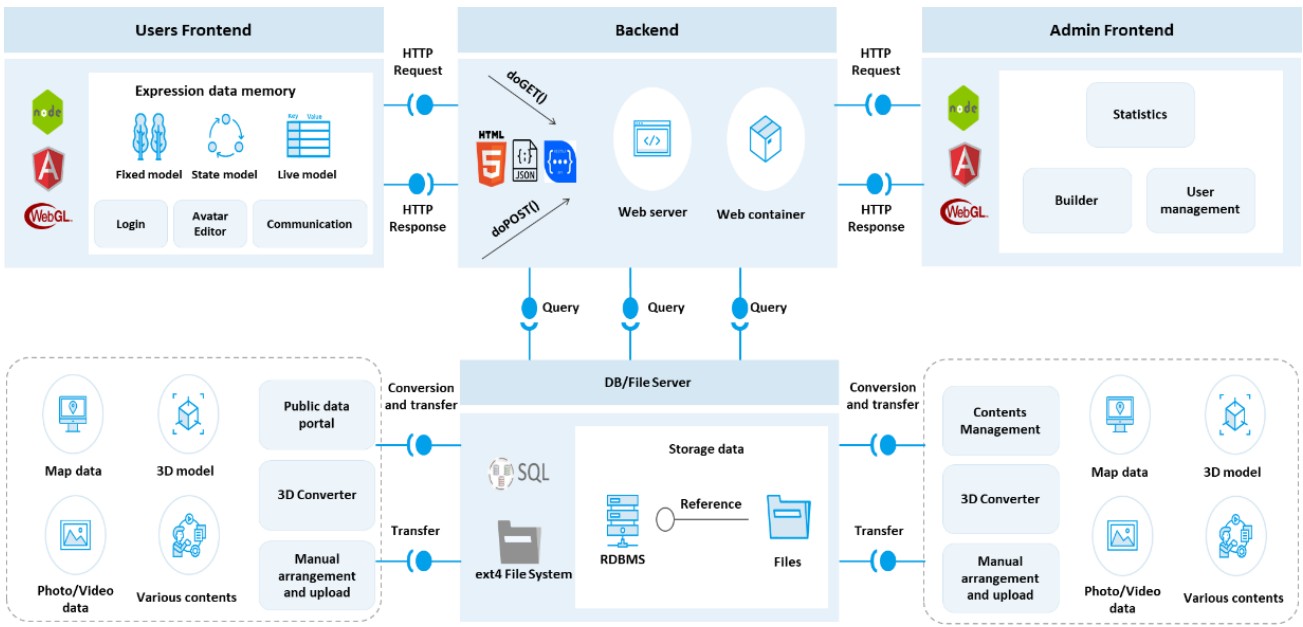

**Figure 1.** System Architecture.

**Table 1.** Standard Technologies.

| Technology | Description |
|---|---|
| RDBMS | - Software to create, modify, and manage relational DB<br>- Store all data on the Metaverse platform (OnTwins) and maintain persistence |
| Ext4 FILE System | - File system used by Linux OS<br>- Store various 3D models, images, and videos |
| HTML 5 | - A next-generation web standard that expresses and provides various applications such as multimedia<br>- Used as all static/dynamic web standards of the platform |
| Angular | - Google's open-source JavaScript framework for SPA development<br>- Used as a frontend development framework |
| NODE | - JavaScript runtime built with JavaScript engine<br>- Used as a platform for all frontend codes |
| WEBGL | - JavaScript and web-based graphics library<br>- Responsible for 3D rendering of the platform |
| RESTful API | - Network-based architecture that can take advantage of the HTTP protocol<br>- Used as a frontend/client basic network standard |
| JSON | - An open format standard for passing data objects<br>- Used as a format of expression data |

### 3.2.2. Server

The eGovFrame [21], developed by a Korean government agency, is a web-based application framework. Based on Java Spring framework and famous Java libraries such as MyBatis, Jackson, and Apache Commons, common functions frequently used in websites such as government public institutions, and public corporations are provided as common components. The eGovFrame consists of development environment, execution environment, operation environment, and management environment. The eGovFrame is a secondary processing framework developed based on Spring Framework. OnTwins has

been developed based on the eGovFrame. It complies with the Public Data Management Guidelines of the Ministry of Public Administration and Security.

OnTwins is basically based on MongoDB, but in this research, Cubid DB [22] is used due to the government operation environment. Large files such as 3D models, videos, and images are managed as a separate file system with reference to the DB. Common data, such as policy data, research data, historical data, map data, and ecological data, related to the Metaverse are acquired from the public data portal system through RESTful API. Collected 3D models are converted through a 3D Converter and periodically uploaded to the platform through a run batch module.

Our own GIS DB, to manage various standards or non-standard map data, have been built. Figure 2 shows an example of UML schema of GIS DB. Considering each different type of metadata and quality characteristic, it normalizes according to the purpose of the map to be serviced, classifies into shape information and attribute information containing various types of map information, and stores in a relational DB. The shape information consists of a 3D terrain model or a 3D CAD model that can be converted into various 2D images, depending on the purpose. The attribute information is designed and implemented to store various types of information such as land type, region classification, buildings, and various map information.

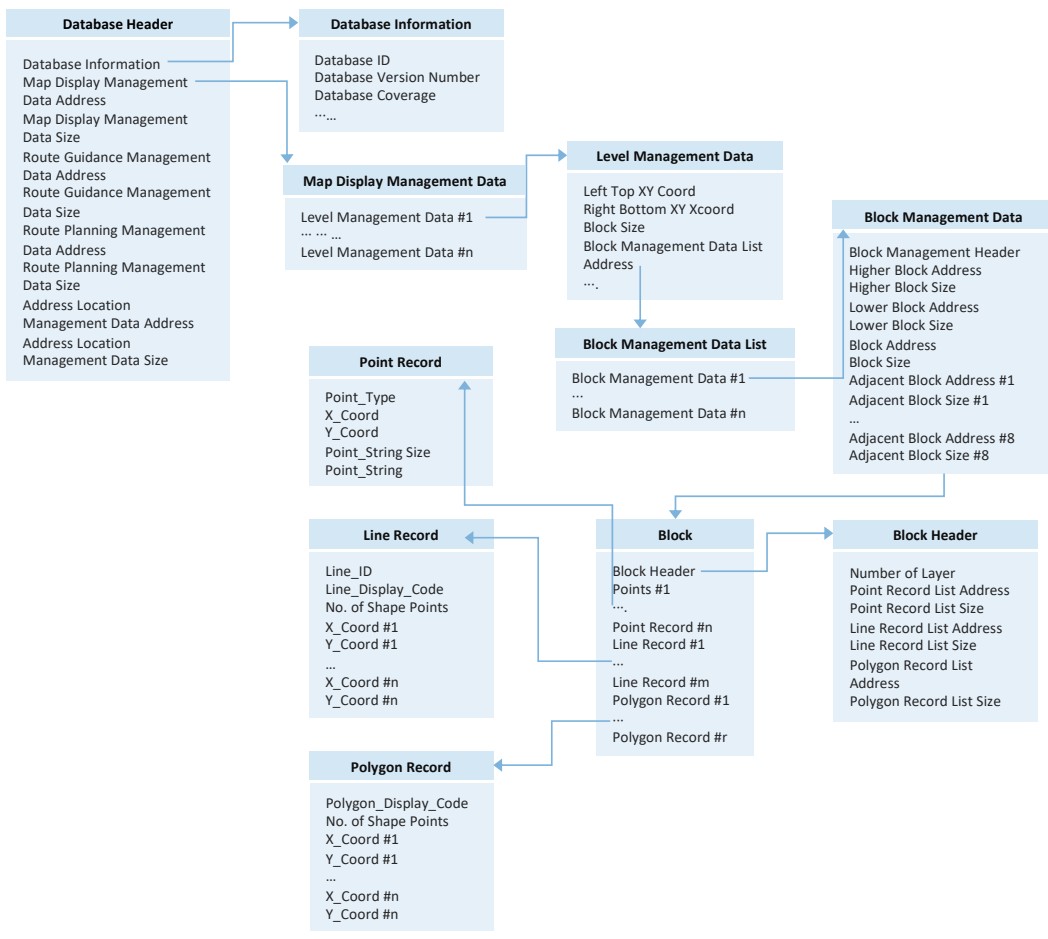

**Figure 2.** UML Class diagram of GIS DB.

The 3D libraries for terrain, natural features, ecological elements, and avatars have been built. The 3D models are produced after confirming the overall range of expression targets and setting the concept. The 3D models are produced by considering the number of meshes, the convenience and scope of using textures, and the optimization of data. There

are 3D modeling tools, such as 3D Max, Blender, and ZBrush, as well as various Visual effects (VFX) tools that are being widely used to create realistic content.

It is equipped with a 3D Converter that converts externally collected or self-produced 3D models to fit the platform. As shown in Figure 3, CAD data, Building Information Modeling (BIM) data such as Industry Foundation Classes (IFC), mesh data, and scan 3D data can be extracted through a CAD kernel, such as ACIS, Parasolid, Open CASCADE, or through APIs provided by each CAD. In the case of CAD data, after extracting the boundary representation structure, the primitive data is classified from the boundary representation structure and used by the recognition module. When mesh data is input, it is analyzed to determine the relevant case among 1, 2, and 3 levels, and after the parameters are extracted, graphic commands are generated and interfaced to Babylon.js, as shown in Table 2 [23].

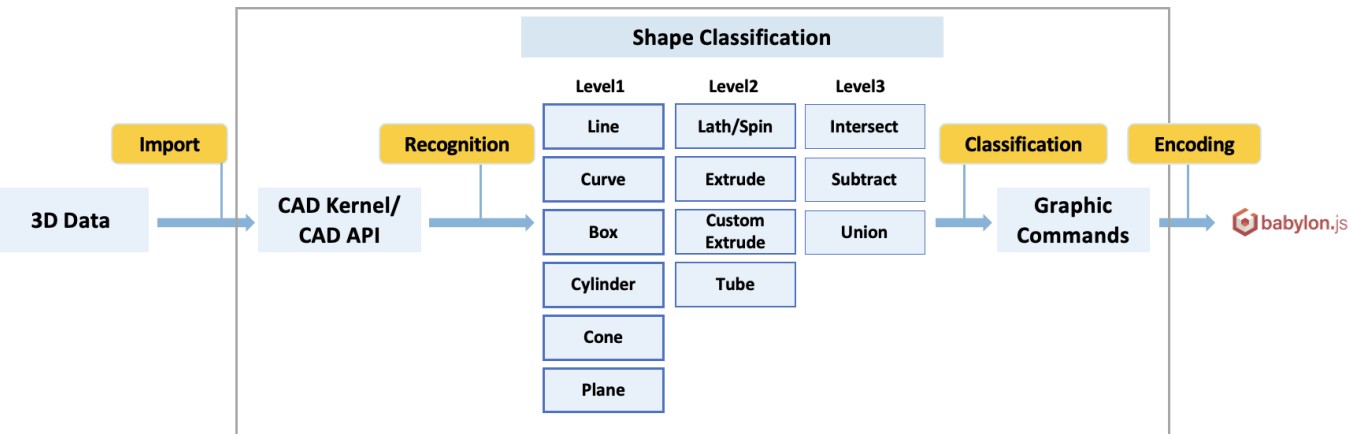

**Figure 3.** The 3DConversion method.

**Table 2.** Mapping graphic commands.

| Level | Primitive/Shape/CSG [1] | Babylon.js |
|:---:|:---:|:---:|
| 1 | Box | CreateBox |
| 1 | Cylinder | CreateCylinder |
| 1 | Cone | CreateCylinder |
| 1 | Sphere | CreateSphere |
| 1 | Plane | CreatePlane |
| 2 | Tube | CreateTube |
| 2 | Extrude | ExtrudeShape |
| 2 | Lathe | CreateLathe |
| 3 | Union | a.union (b) |
| 3 | Intersect | a.intersect (b) |
| 3 | Subtract | a.subtract (b) |

[1] CSG: Constructive Solid Geometry.

The OnTwins data models are divided into a storage data model that is saved as a DB and file in the form of persistence, as well as an expression data model that is used in memory for effective information transfer and processing in the real system. The data model of the Metaverse requires efficient storage of various types of information from numerous users. Therefore, the data model has been developed based on the four characteristics related to transactions: atomicity, consistency, isolation, and durability for persistent data storage and efficient management. The storage model has two types: structured data stored in relational DB format and unstructured data stored in the file system, as shown in Figure 4. The structured data consists of the Metaverse spatial models such as terrain and features, the Metaverse public state data such as avatar state, feature state, and state history by time, the user data such as user-to-user and system-to-system exchanges, and the content model. The unstructured data include 3D library model, photos, video data,

and content data. Figure 5 shows the DB design for stored data. Table 3 is a description of each entity.

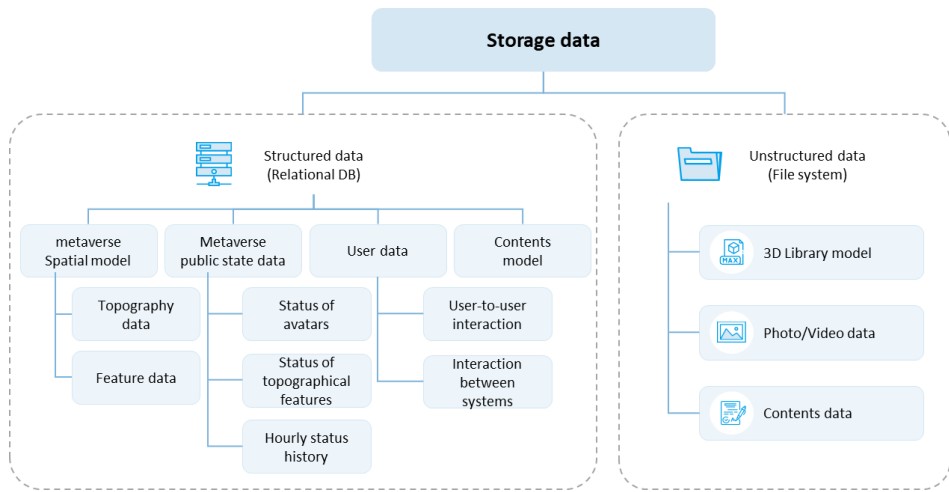

**Figure 4.** Storage data model.

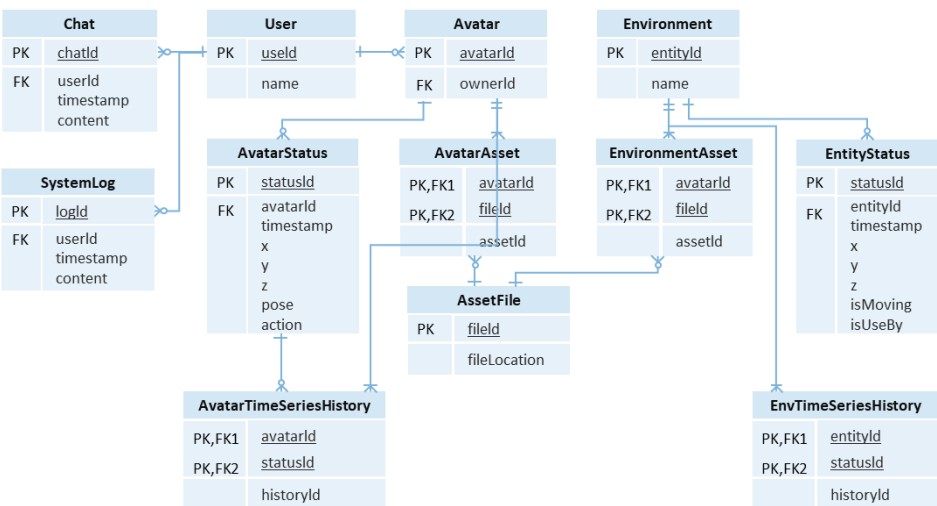

**Figure 5.** Storage data model entity.

**Table 3.** Description of Each Entity.

| Entity | Description |
|---|---|
| User | - User data |
| Avatar | - User's avatar data |
| AvatarStatus | - State data of user avatar <br> - Possible to search history as time is recorded together |
| AvatarAsset | - Asset file data representing user avatar |
| AssetFile | - Files stored in an external file system |
| Environment | - Environment and feature data |
| EntityStatus | - State data of environment and features |
| EnvironmentAsset | - Asset file data representing environment and features |
| AvatarTimeSeriesHistory, EnvTimeSeriesHistory | - For time series data inquiry |
| Chat, SystemLog | - Exchange data between users with the system |

### 3.2.3. Client

The stored data model is not suitable for information transfer and rendering processing. Therefore, the expression data model is designed as shown in Figure 6. The expression data has been implemented based on the following three design principles. The first is the efficiency of composing data, which allows it to be used as efficiently as possible by eliminating data duplication through the reference ID method. The second is a hierarchical tree structure, which reduces the amount of data required for communication and speeds up operations. Lastly, the third is variability that can be flexibly applied as the expression data is variably configured according to the user environment. The expression models are divided into the fixed model, the state model, and the live model. The fixed model is not changing, unlike the Metaverse public state data, the user data, the contents model, and the history log. The state model is changing frequently just as the character's current position, posture, and movement data. The live model includes the user interaction and the notice information. The expression data model is based on the JSON model for efficient rendering. It is expressed in a tree structure suitable for communication and search.

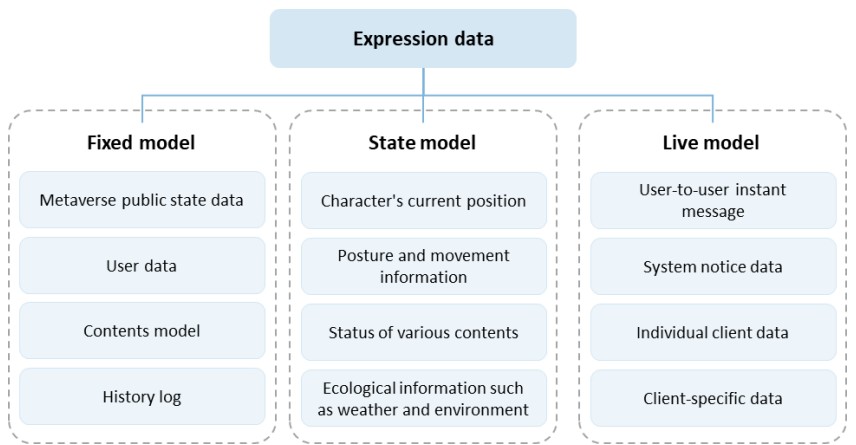

**Figure 6.** Expression data model.

Hybrid App means an app that can be developed once and can be used on various platforms such as iOS, Android, and Windows. It is advantageous in terms of cost reduction and development speed. A Hybrid App, such as Native App, is distributed through the App Store and Play Store, can control the hardware of mobile devices, and has the advantage of faster app loading and execution than the mobile web. Since The OnTwins mobile has been developed in the form of an HTML5 web app, it can be quickly converted to a mobile app. OnTwins also supports various VR devices, such as Window Mixed Reality and Oculus. It is possible to experience a panoramic view through the fly through support, and it is easy to understand the current location when walking through the mini map. In addition, the sense of reality is increased by supporting high-quality sound. It also supports 3D model visualization on AR-based on markers.

Various algorithms have been applied for the optimal visualization of large-capacity virtual space. First, an out-of-core algorithm that uses the system memory instead when the GPU has insufficient memory is applied. It is possible to visualize a 3D scene with a capacity larger than the memory of the user's computer. Second, the occlusion culling technique, which reduces graphic memory consumption by excluding the dotted line model of the conceptual diagram, is implemented. The applied occlusion culling technique pre-processes a 3D scene into a tree structure and a graph structure to efficiently determine and visualize objects that need visualization at runtime, thereby maximizing the visualization speed. Third, it is equipped with the Physically Based Rendering (PBR) technology. By providing a more realistic 3D visualization, the immersion of the Metaverse system has been maximized. Fourth, Level of Detail (LOD), a technology that adjusts the precision of mesh

modeling data in stages, is applied. By adjusting the number of polygons when rendering terrain or objects, high quality is expressed without the degradation of image quality.

Communication functions based on VoIP, I18n, and Text to Speech (TTS) are implemented to OnTwins. The existing telephone method is a method of making a mutual call with a phone number already assigned in a fixed place. The IP Telephony method, which appeared with the introduction of VoIP technology, allows users to make calls with their own numbers anywhere, regardless of time or place. It means that the IP address and user information are converted into a DB and are interconnected through this. I18n refers to the process of designing and developing internationally accepted software so that the software does not depend on a specific region or language environment. OnTwins supports multiple languages, using i18next, based on the string substitution method. In addition, a translation function is supported by using a transformer, which is a natural language processing model. In the case of interpretation, it is converted into Speech to Text (STT) and translated, then converted into TTS and delivered, as shown in Figure 7. TTS is a technology that automatically generates sound waves of speech sounds. The voice of a person selected as a model is recorded and divided into certain voice units. Then, the code is put into a synthesizer and, according to the instructions, only the necessary phonetic units are put back together to artificially create speech sounds. OnTwins utilizes the AI TTS system developed by Facebook.

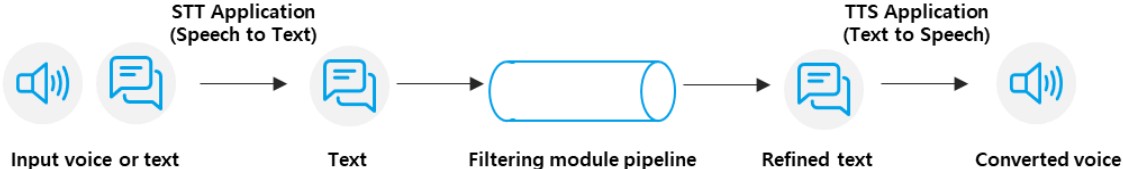

**Figure 7.** Communication method.

## 4. An Application Case

### 4.1. Purpose

After the Korean War, the DMZ was decided in an area of 2 km from north to south (total of 4 km) centered on the Military Demarcation Line (MDL). Although military activity is prohibited, it is the place where the fiercest confrontation takes place because it is the border area between South and North Korea. In the DMZ, both sides build barriers such as barbed wire and conduct thorough surveillance and espionage against each other. They promote each other's regimes through broadcasting, and even sometimes armed conflicts occur in the DMZ. The activities of the public are strictly restricted. The natural ecosystem is preserved, and it is attracting attention as the largest nature reserve in Asia. According to a survey report released by the National Institute of Ecology, the DMZ is home to 5929 species of wildlife, including 101 endangered species. The high attention from Korean people, as well as from people around the world, can be a great help for easing tensions and establishing peace in the DMZ. Therefore, the Korean government is actively developing peace policies such as public visits and walking to the DMZ events. Building a Metaverse can also be a good peace policy. The DMZ, where the public access is restricted, is a highly appropriate area for the Metaverse. The ultimate goal is to create a space where people from all over the world can visit the DMZ at any time, from anywhere so that the DMZ Metaverse can draw the special attention of the young generation and people all over the world to further establish peace.

### 4.2. Public Service

As shown in Figure 8a, the Metaverse public service is operating stably in the government software operating environment. The site name "DMZ Universe" is a combination of two keywords, DMZ and universe. The slogan "My New Universe" is to convey the meaning of returning the DMZ, which people are not easily allowed to visit, to the public

through the Metaverse. It has the meaning of creating one's own new universe (world) in the DMZ Metaverse space. As shown in Figure 8b, the logo of DMZ Universe is a design that utilizes the meaning of universe by using the shape of a planet. It has the meaning of social by shaping the planet in the shape of a speech bubble. It is not simply a web system, but it contains the meaning of a new world where there is interaction with users.

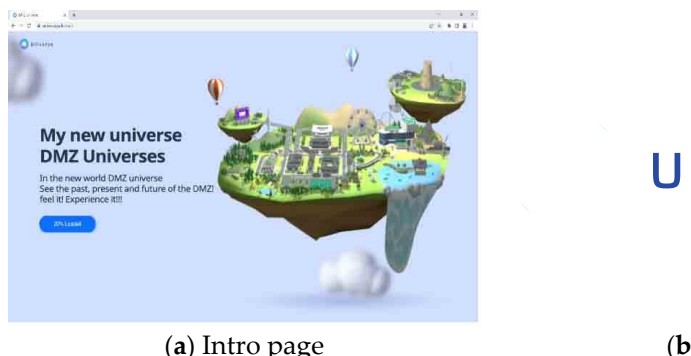

(**a**) Intro page　　　　　　　　　　　　　　(**b**) Logo

**Figure 8.** DMZ Universe (www.universe.go.kr accessed on 5 August 2022).

In the first virtual space where users enter, there are three islands: Peace Island, Woori Island, and Sky Island, as shown in Figure 9. The buildings on each island are related to the DMZ or designed with the motif of Korean landmarks. The labels on the island are the sitemap of the DMZ Universe, and users enter the public functional space by clicking the space of each island or clicking the label.

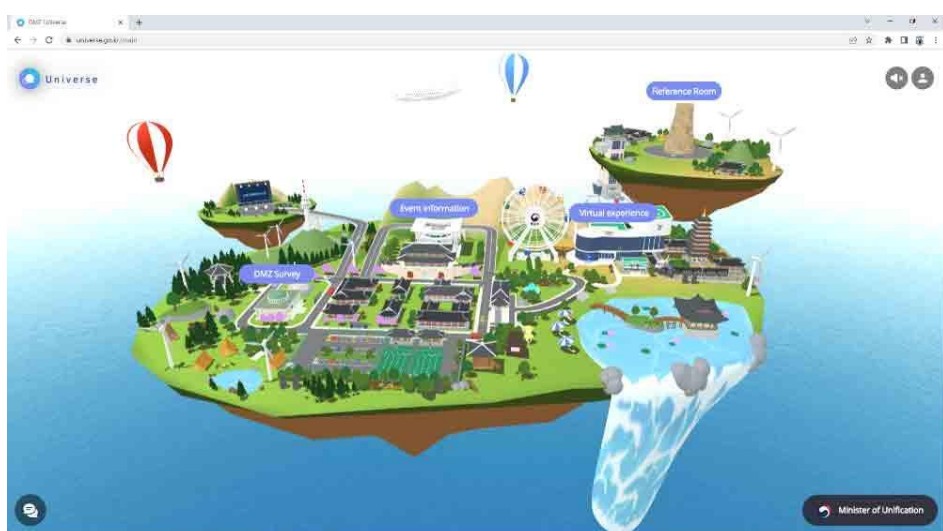

**Figure 9.** Landing Page for virtual space.

As shown in Figure 10a, the resource center is designed with the motif of *Cheomseong-dae*, Korea's first astronomical observation facility. Each floor of *Cheomseongdae* made of stone is rotating, and it is designed with the intention of delivering a paranormal appearance to the user. As shown in Figure 10b, DMZ-related humanities and scientific materials are provided in the internal space of the resource center. In particular, the users can visually check the DMZ vegetation, culture, and history through the 3D simulator.

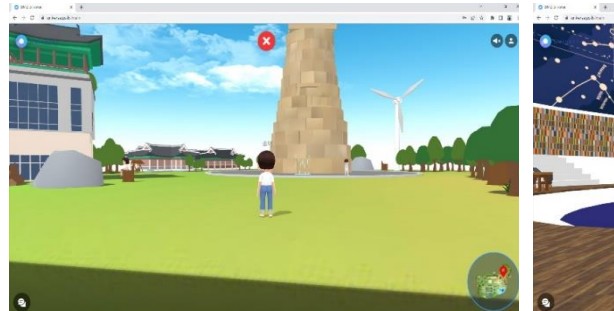
(**a**) *Cheomseongdae*

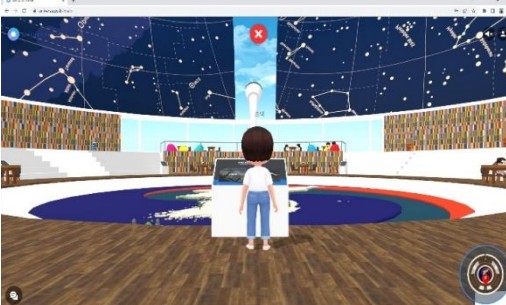
(**b**) Inside of Resource Center

**Figure 10.** Resource Center.

As shown in Figure 11a, the DMZ survey space is designed with the motif of the Korean National Assembly building. Users can check the actual condition data of the DMZ that has been accumulated for decades in this space as if looking in an exhibition, as shown in Figure 11b. The data is newly updated on the platform every quarter.

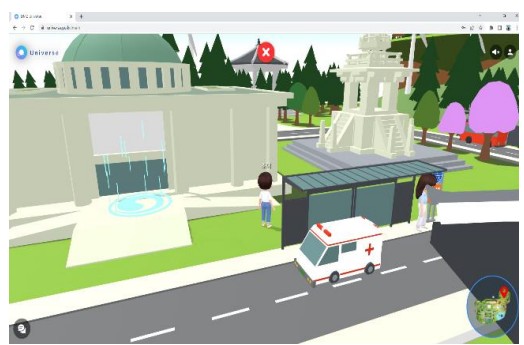
(**a**) Korean National Assembly Building

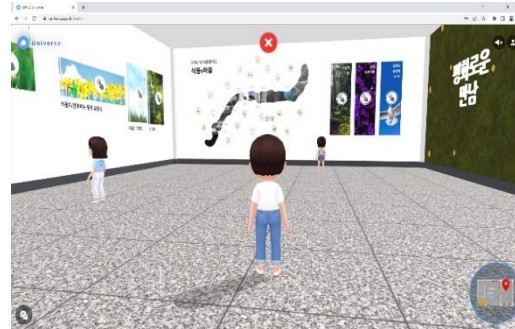
(**b**) DMZ Data Center

**Figure 11.** DMZ Actual Condition Data.

As shown in Figure 12a, there is a publicity screen connected to an external link on Woori Island. Websites related to the Ministry of Unification appear according to time, and by clicking on the screen, it is linked to a related page. When the user clicks the image of the Peace map on the display, as shown in Figure 12b, it is also connected to the Peace map. The Peace map is a web map that provides a total of 12,000 pieces of information related to the DMZ. Information on the geographical topography, historical culture, ecological environment, and unification and peace of the North–South border area is included. The DMZ web map is created for the purpose of helping various educational sites, tourism, and policy establishment.

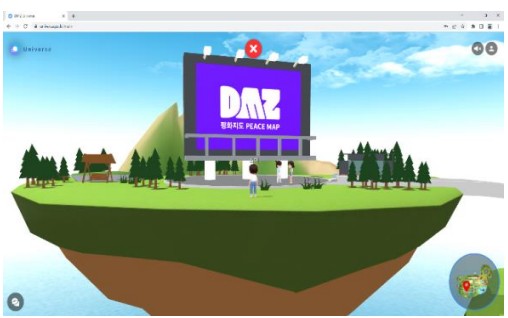
(**a**) Woori Island Publicity Screen

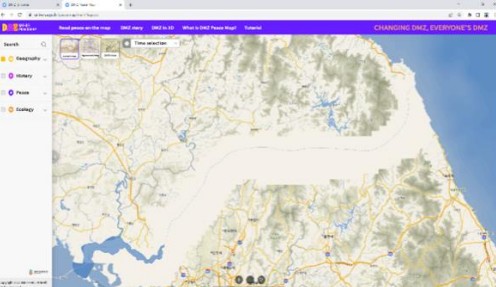
(**b**) Peace Map

**Figure 12.** Public Relations Space.

Users enter the event information space, as shown in Figure 13b, through *Namdaemoon*, as shown in Figure 13a. The event information space is designed with the motif of the House of Freedom in Panmunjom. It is a space where various events held by the Ministry of Unification are promoted, and users can apply for the events. The user can acquire various types of event information by walking around the event information space. In addition, a friendly interactive civil service via a chatbot is provided. Every year, information on major events, such as the Walk of Peace and the DMZ Peace Forum, is provided.

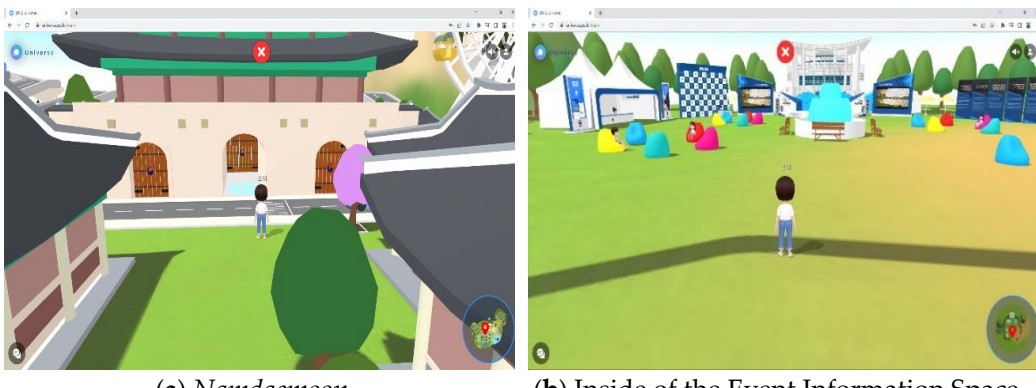

(**a**) *Namdaemoon*      (**b**) Inside of the Event Information Space

**Figure 13.** Event Information.

As shown in Figure 14a, Peace Station of the DMZ virtual experience is designed with the motif of Seoul Station. As shown in Figure 14b, there are booths inside Peace Station that allow users to travel to various times and spaces, such as *Goseong* in 2022, *Cheorwon* in 1930, and Paju in 1943. Users can experience traveling through time and space by train, as shown in Figure 14c. The waiting area at Peace Station is used as a guide for various events and missions. In particular, it is a space that provides the Generation MZ with the history and meaning of unification in an easier and friendlier way.

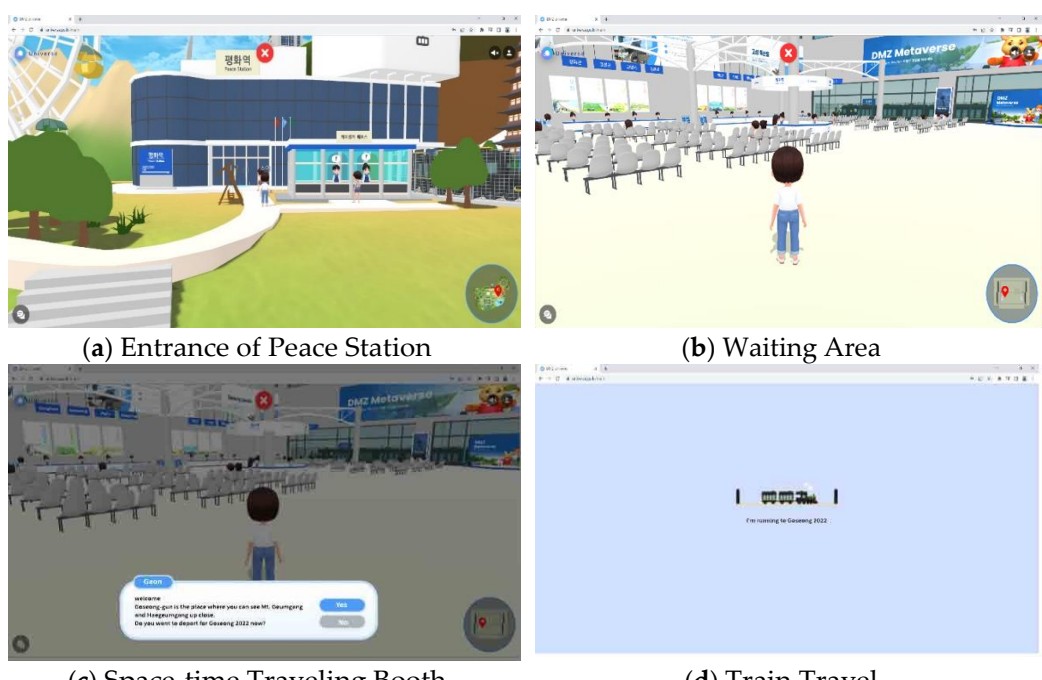

(**a**) Entrance of Peace Station      (**b**) Waiting Area

(**c**) Space-time Traveling Booth      (**d**) Train Travel

**Figure 14.** Peace Station.

As shown in Figure 15a, *Goseong* 2022 is a virtual space created based on the actual *Goseong* in Gangwon-do. As shown in Figure 15b, *Goseong* Unification Observatory and Mt. *Geumgang* Observatory, as well as various DMZ facilities are designed. Various posters of elementary school students expressing the hope of peace and unification are displayed on the bordered wire fence, as shown in Figure 15c. The user's comments received through SNS are posted on the wish tree for one hour. As shown in Figure 15d, users can see the photos of real *Goseong* through the telescope in front of the Unification Observatory.

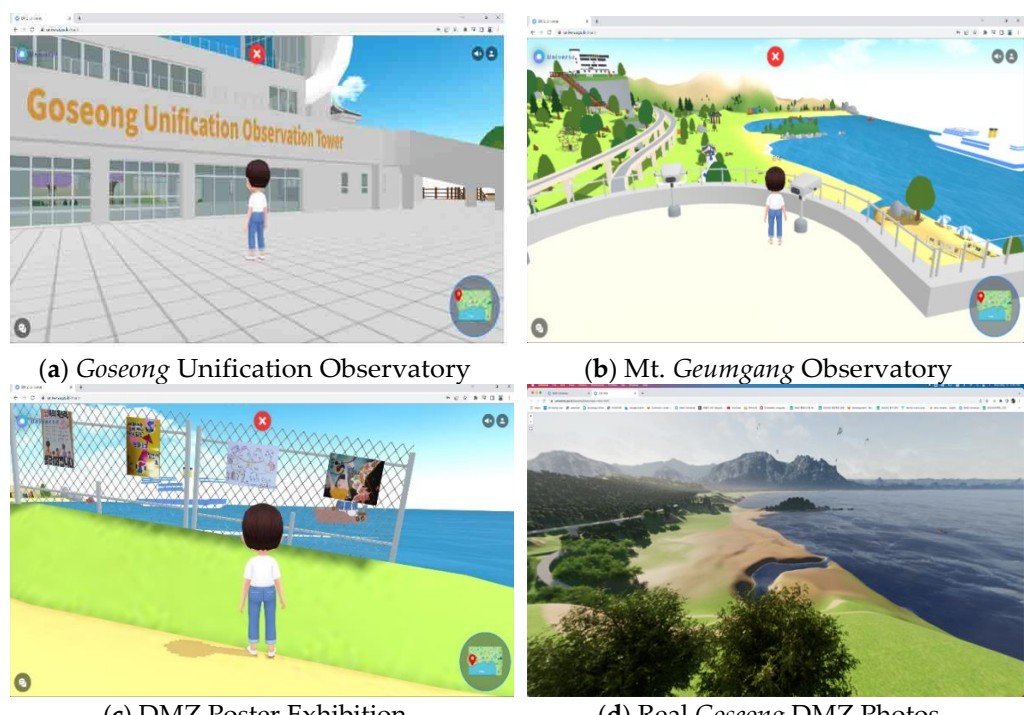

(**a**) *Goseong* Unification Observatory (**b**) Mt. *Geumgang* Observatory

(**c**) DMZ Poster Exhibition (**d**) Real *Goseong* DMZ Photos

**Figure 15.** *Goseong* DMZ.

It is possible to create avatars that bring out the individuality of the user. The created avatar is used as a profile image in the DMZ Universe. Users can change the hairstyle as shown in Figure 16a. Users also can create avatars that fit their personal tastes, as traditional and modern clothes are provided to choose, as shown in Figure 16b. Differentiated cute characters, unique to the DMZ Universe, are designed to give friendly feelings while harmonizing with the Ministry of Unification Metaverse.

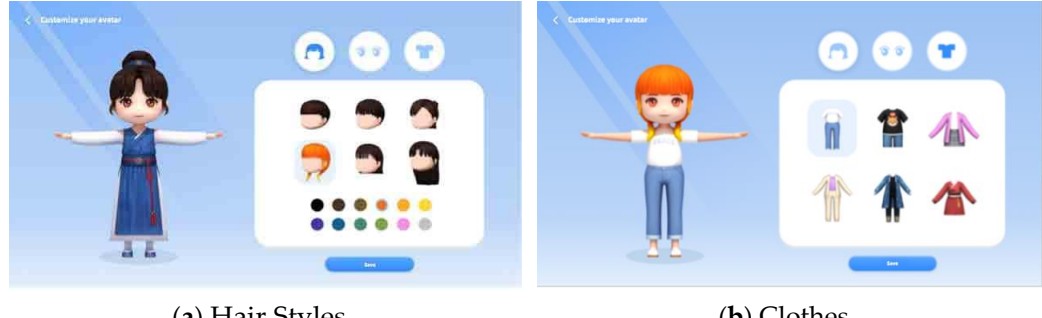

(**a**) Hair Styles (**b**) Clothes

**Figure 16.** Customizing Avatar.

Finally, users can communicate via text and voice chat. Friend addition and management functions are provided. All chat rooms are provided, and group chat rooms can be created. Voice chat is only available 1:1.

### 4.3. Discussion

There are seven necessary features related to being a sustainable Metaverse platform. The seven features are: (1) persistence that the Metaverse continuously exists regardless of whether the user is connected, (2) mass concurrency, where many things happen in real time, (3) unrestraint that anyone can participate at the same time, (4) an economic system that can secure the same economic system as reality, (5) transcendence connecting the real world and the online space, (6) content for which individuals and companies can provide various contents, and (7) interoperability that can be exchanged between Metaverse platforms.

The DMZ Universe in the basis of the aforementioned seven features is discussed. First, there is no pause, reset, or end of the DMZ Universe. Users can experience the DMZ space anytime, anywhere by accessing the URL without any additional installation. Second, in this Metaverse space, many things happen vividly in real time, just as in reality. Users can take pictures while taking a walk in the DMZ Universe with their friends and improvising games, such as hide and seek. Third, the OnTwins platform uses client resources. Therefore, it supports a system architecture in which the load on the server is less, and numerous users can participate at the same time. Fourth, the OnTwins platform is designed to enable economic activities, such as sales, ownership, and investment by individuals and companies, and its functions are being implemented. However, as DMZ Universe is a government public service, there are restrictions on paid services. Only basic avatar accessories will be provided free of charge, and a service will be launched to advertise and promote the sale of local delicacies in the DMZ area. Fifth, it is possible to travel not only the present time but also various times and spaces in the DMZ Universe. Various spatiotemporal contents can be loaded on the OnTwins platform. Figure 17 shows that the *Cheorwon* and Sinuiju areas in the 1930s were restored as Metaverse spaces and prepared for service. It was built by reflecting the realities of cattle markets, running races introduced in newspapers, kindergartens, and local markets at that time.

Sixth, various services are possible by providing rich content by individuals and companies. Contents created with commercial 3D modeling software can be loaded according to the guidelines of OnTwins. Cases in *Cheorwon* and *Shinuiju*, shown in Figure 17, are examples. In the future, our plan is to provide user-friendly UI and functions to add content more conveniently. Seventh and last, exchanges such as teleportation between Metaverse platforms should be possible. Currently, the technical hurdles are very high in this part, and support is currently unavailable. In order to support heterogeneous Metaverse platforms such as Roblox, Zepeto, and Gather, the development of a data interface standard that can be mapped with the data model between each platform must be preceded. However, the data model of each platform is a core technology that cannot be disclosed. Therefore, development of a standard neutral format, such as STEP of CAD, is preceded, and interoperability will be secured only when this standard model is supported on all platforms. However, it will be difficult for any platform to support interoperability, realistically, in the near future because it takes a long time to develop and participate in standards, and a great deal of trial and error is required to implement them in an actual system. Table 4 shows comparison with other platforms based on the above-mentioned seven features.

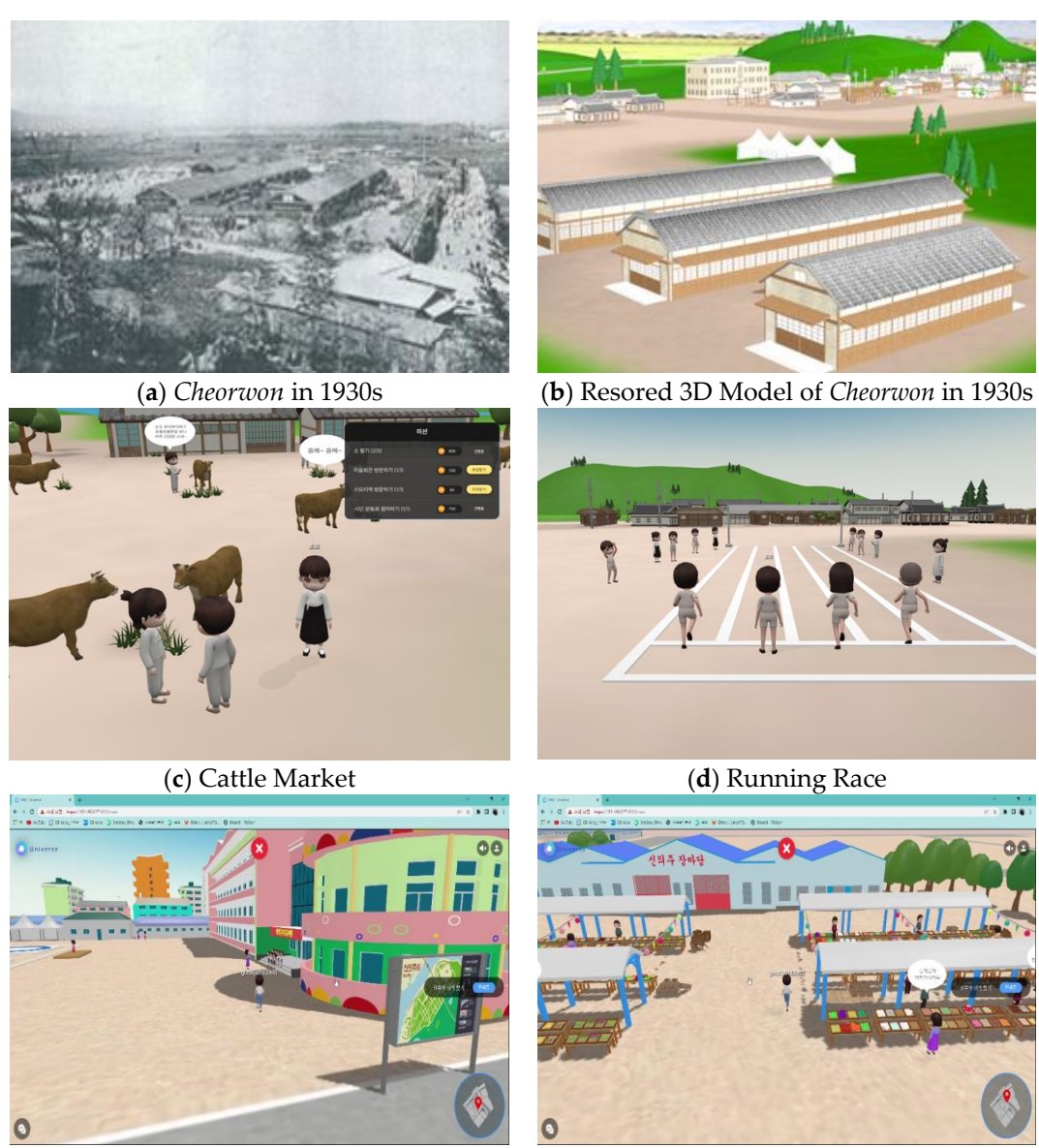

(**a**) *Cheorwon* in 1930s　　(**b**) Resored 3D Model of *Cheorwon* in 1930s

(**c**) Cattle Market　　(**d**) Running Race

(**e**) Kindergarten in *Shinuiju*　　(**f**) *Shinuiju* Local Market

**Figure 17.** Past *Sinuiju* and *Cheorwon*.

**Table 4.** Compare Metaverse Platforms based on Seven Necessary Features.

|  | Roblox | ZEPETO | OnTwins |
| --- | --- | --- | --- |
| **Persistence** | O | O | O |
| **Mass concurrency** | O | O | O |
| **Unrestraint** | △ | X | △ |
| **Economy system** | O | O | △ |
| **Transcendence** | High | High | High |
| **Content** | High | High | Low |
| **interoperability** | X | X | X |

Roblox, ZEPETO, and OnTwins all support the Persistence factor that exists continuously, regardless of whether the user is connected or not, as well as the Mass concurrency factor that occurs in real time. Regarding the Unrestraint, ZEPETO is limited to 16 participants per each space. Theoretically, Roblox and OnTwins have no limit on the number of participants, but performance may deteriorate as the number of users increases, due to the limitations of the hardware environment. Therefore, since it cannot be said that this

factor is completely satisfied, it is marked with a triangle. Roblox and ZEPETO support the Economy system, and OnTwins has the function, but the DMZ Universe does not currently provide the service. All three Metaverse platforms satisfy the factor of Transcendence that connects the real world and the online space. Roblox and ZEPETO support the Content factor that allows individuals and companies to provide various contents. Relatively, OnTwins has its function, but it is weak compared to other platforms. Neither platform currently supports interchangeable interoperability between Metaverse platforms.

There is also another feature that the Metaverse should be composed of the three basic elements of 3D space: Avatar, Activity, as well as the 5Cs—Canon, Creator, Currency, Continuity, and Connectivity—which are unique features that differentiate it from other services [17]. The Metaverse platform developed through this research satisfies all 5Cs, as described above, but only the Currency part cannot be provided due to government public services. Table 5 compares the system requirements of Roblox, Zepeto, and OnTwins. Although it has relatively low system requirements, OnTwins has the advantage that it is free to access the OS and does not require a separate installation.

**Table 5.** Compare System Requirements.

|  | **Roblox** | **ZEPETO** | **OnTwins** |
|---|---|---|---|
| **OS** | Windows 7, 8, 8.1, 10 OS X 10.7 | Windows 10 or above Mac OS Mojave or above | No OS requirements as it is served via browser |
| **CPU** | 1.6GHz or above (latest processor) | Intel i5 or above | Intel i5 or above |
| **MEMORY** | 1GB or above | 8GB RAM or above | 8 GB RAM or above |
| **Graphic** | DirectX Ver. 9 above and Shader Model 2.0 | Geforce GTX 660 or above, DirectX Version 10 or above | GeForce GTX660/Intel HD Graphics 4000 |
| **HDD Space** | 20 MB or above | 500 MB or above | No separate installation space required |
| **Internet** | 4–8 Mb/s |  | 4–8 Mb/s |

Table 6 shows a comparison of the pros and cons of each platform. Roblox has a high degree of freedom ad many users, Zepeto has the advantage of custom functions, and OnTwins has the advantage of free extensibility. Roblox has relatively low character quality, Zepeto has a limit of 16 users, and OnTwins is a new system, so recognition and instability are considered as its disadvantages.

**Table 6.** Compare Seven Features.

|  | **Roblox** | **ZEPETO** | **OnTwins** |
|---|---|---|---|
| Pros | - Game-oriented arcade<br>- Large number of users<br>- High degree of freedom | - Specialized custom functions<br>- Large number of users | - Free scalability<br>- Technical support |
| Cons | - Relatively low character quality<br>- Technical support | - 16 people limit<br>- Low degree of freedom<br>- Restrictions on content provision | - Lack of function<br>- Initial system instability<br>- Lack of awareness |

## 5. Conclusions

Metaverse is a technology that allows users to experience a sense of presence and immersion in a space that is difficult to visit. Vicarious pleasure can keep users interested. The Korean government has been conducting public events using the commercialized Metaverse platforms. However, for sustainable operation, they had no choice but to rely on commercial solutions from external companies, and there was always a risk of not being

able to control cost and security issues on their own. Therefore, it was absolutely necessary to introduce a Metaverse platform that operates in the government cloud operating environment, based on the government IT framework.

In this paper, the Metaverse platform, developed based on the government IT framework and operated in the government cloud environment, was described in detail. In addition, the case of DMZ Universe built for the Korean DMZ was explained, compared, and discussed with other Metaverse platforms. The OnTwins platform has the advantage that it can be used by accessing the URL without additional installation, as it is developed based on the web. Currently, it is stably operating as the first Metaverse platform of a public institution in the Korean government. Since there have only been a few studies on Metaverse platform development so far, this research will be a good reference.

There is quite positive feedback about being able to experience the difficult-to-visit DMZ area with the Metaverse itself. Feedback was already collected about the need for a variety of content that allows users to continuously revisit and stay in this space. Various games, such as OX quiz, scavenger hunt, hide and seek, tag, and archery will be equipped within this year to further provoke user interest. It is planned to encourage elementary school students to come to the DMZ Metaverse for field trips and increase their interest in and understanding of the Korean War and peace through games and experiences.

It is planned to open a mobile environment support service and provide a user-friendly UI where individuals and companies can conveniently add various contents. Moreover, it will help revitalize the DMZ regional economy by linking the multilingual service and local delicacy sales function. A personalization function will also be added, so users can decorate their own space. Finally, a quantitative and qualitative evaluation of the Metaverse platform usability will be carried out.

**Author Contributions:** Conceptualization, S.C. and J.W.; methodology, S.C. and J.W.; software, K.Y., J.Y., M.K., B.L. and I.S.; validation, K.Y., J.Y., M.K. and B.L.; investigation, K.Y. and M.K.; writing original draft preparation, S.C.; writing review and editing, S.C., K.Y., J.Y., M.K., B.L., I.S. and J.W.; visualization, M.K.; supervision, J.W.; project administration, S.C. and K.Y. All authors have read and agreed to the published version of the manuscript.

**Funding:** This research received no external funding.

**Institutional Review Board Statement:** Not applicable.

**Informed Consent Statement:** Not applicable.

**Data Availability Statement:** Not applicable.

**Acknowledgments:** We are especially grateful to Ministry of Unification.

**Conflicts of Interest:** The authors declare no conflict of interest.

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
