# Peer review of "Building Korean DMZ Metaverse Using a Web-Based Metaverse Platform"

_applsci, doi:10.3390/app12157908_

Round 1

Reviewer 1 Report

I found some little mistakes in the text and I upload a file with highlighted words. The formatting needs improvement, in some places first line indent is needed  (paragraphs 3.1.1-3.1.7) . 

The platform OnTwins is very well decribed in details with diagrams and a lot of pictures.

Is there an English version of this platform? As I see everything is in Korean language only.

Author Response

I found some little mistakes in the text and I upload a file with highlight words. The formatting needs improvement, in some places first line indent is needed. (paragraphs 3.1.1-3.1.7) 

-> Thank you for kinds comments. We did improve the formatting using bullets

The platform OnTwins is very well described in detail with diagrams and a lot of pictures.

-> We appreciate your comment. We wrote the paper in detail so that it can be a good reference when developing Metaverse platform

Is there an English version of this platform? As I see everything is in Korea language only.

-> Sorry for the inconvenience. English version will be available within this year. 

Reviewer 2 Report

Manuscript ID: applsci-1821056

Building Korean DMZ Metaverse Using a Web-based Metaverse Platform

The authors present the Metaverse platform based on the government IT frame work and operated in the government cloud environment.  The topic covers the case of DMZ Universe built for the Korean DMZ and compared to other Metaverse platforms. The development is explained in great detail.

The paper is interesting as it shows an example of a possible metaverse, however the scientific contributions are unclear.

The authors could consider the following suggestions for improvement.

11.     Paper contributions should be clearly explained in more detail. The scientific contributions / aims of the paper are unclear.

22.     The authors should elaborate, if possible, on any user evaluations or feedback given.

33.    The authors should explain what symbols and features in table 4 represent.

Author Response

Paper contributions should be clearly explained in more detail. The scientific contributions / aims of the paper are unclear.

-> Most of the studies related to the Metaverse are studies that analyze the effect of application in specific fields such as education using a commercial Metaverse platform. Also, there are many studies on detailed techniques and algorithms related to the Metaverse and games. There is very little research on Metaverse platform development itself. This paper introduces the system architecture, model, and functions related to Metaverse platform development. Researchers (readers) are expected to be able to understand the overall development method for Metaverse platform development and refer to this paper when developing related systems and functions. Section 2 and 5 have been updated. 

The authors should elaborate, if possible, on any user evaluations or feedback given

-> There was a lot of positive feedback about experiencing the DMZ, which is difficult to visit, through the Metaverse as it was introduced in the Korea media. There was feedback that there was still a lack of elements that could provide users fun. This section has been added to section 5. Currently, various functions such as games are being added. We will carry out a quantitative and qualitative assessment of the Metaverse platform and it mentioned as a future work in Section 5.  

The authors should explain what symbols and features in table4 represent.

->For clearer understanding, the title of Table 4 has been revised. The first part of the 4.3 Discussion (line 507 – 551) describes the 7 functions that a sustainable Metaverse platform should have. 

Round 2

Reviewer 2 Report

Th authors have somewhat addressed the issues raised. Some explanation should still be given to table 4, as it is still unclear, what specific symbols in the Table 4 represent (circle, triangle, cross). The authors should provide the necessary explanation.

Author Response

Th authors have somewhat addressed the issues raised. Some explanation should still be given to table 4, as it is still unclear, what specific symbols in the Table 4 represent (circle, triangle, cross). The authors should provide the necessary explanation.

--> Thanks you for you comments. We have added the explanation for Table 4. Please see the line 566 ~ 579